# Anomaly-Based Fault Detection in Wind Turbine Main Bearings

Lorena Campoverde-Vilela[1], María del Cisne Feijóo[1], Yolanda Vidal[2,3], José Sampietro[4], and Christian Tutivén[1]

[1] ESPOL Polytechnic University, Escuela Superior Politécnica del Litoral, Faculty of Mechanical Engineering and Production Science (FIMCP), Mechatronic Engineering, Campus Gustavo Galindo Km. 30.5 Vía Perimetral, P.O. Box 09-01-5863, Guayaquil, Ecuador.
[2] Control, Data, and Artificial Intelligence (CoDAlab), Department of Mathematics, Escola d'Enginyeria de Barcelona Est (EEBE), Universitat Politècnica de Catalunya (UPC), Campus Diagonal-Besós (CDB), Eduard Maristany, 16, 08019 Barcelona, Spain.
[3] Institute of Mathematics (IMTech), Universitat Politècnica de Catalunya (UPC), Pau Gargallo 14, 08028 Barcelona, Spain.
[4] Universidad Ecotec, Km. 13.5 Samborondón, Samborondón, EC092302 Ecuador.

**Correspondence:** Yolanda Vidal (yolanda.vidal@upc.edu)

**Abstract.** Renewable energy is a clean and inexhaustible source of energy, so every year interest in the study and the search for improvements in production increases. Wind energy is one of the most used and therefore the need for predictive maintenance management to guarantee the reliableness and operability of each of the wind turbines has become a great study opportunity. In this work, a fault detection system is developed by applying an anomaly detector based on principal component analysis (PCA), in order to state early warnings of possible faults in the main bearing. For the development of the model, SCADA data from a wind park in operation are utilized. The results obtained allow detection of failures even months before the fatal breakdown occurs. This model requires (to be constructed) only the use of healthy SCADA data, without the need to obtain the fault history or install additional equipment or sensors that require greater investment. In conclusion, this proposed strategy provides a tool for the planning and execution of predictive maintenance within wind parks.

## 1 Introduction

High consumption of fossil fuels increases environmental pollution and increases the effects of climate change. For this reason, it is important to deepen the study and development of renewable energies to achieve a green energy economy (Baloch et al., 2022; May, 2017). Some authors, such as Yii and Geetha (2017) and Kang et al. (2020), focus their research on the fact that technological innovation reduces environmental

degradation and reduces the effects of pollution. It can be seen from the documents of the International Energy Agency of 2019, that the amount of renewable energy as a supply of the OECD (Organization for Economic Cooperation and Development) reached 10.8 % of its primary energy. Similarly, renewable energy is expected to provide one third of total energy generation by 2035. This growth motivates the study and implementation of new technologies and algorithms for their development (Wang, 2022).

One of the topics that has caught the interest of many researchers is the development of new technologies that allow for improved maintenance management in wind turbines (WTs), since a good maintenance strategy is the basis for supporting production with reliability, safety, environmental preservation, and adequate costs. In addition, it guarantees continuous operation and avoids economic losses due to emergency stops or equipment damage (Velmurugan and Dhingra, 2015).

According to May et al. (2015), operating and maintenance costs regularly represent 30 % of the total cost of industrial plants, so poor management or maintenance generates additional costs that can significantly affect their operation and economy. Currently, there are different maintenance strategies, such as corrective, preventive, predictive, and proactive (Bahar et al., 2021; Velmurugan and Dhingra, 2015). In general, wind parks use preventive and corrective maintenance; however, considering industrial environments, where stops can cause large losses (Hu and Chen, 2020), predictive maintenance is becoming a method of growing interest, especially with its new approach geared toward the use of industry 4.0 standards. This new technology has allowed integration between physical and digital systems, facilitating the collection of a large amount of data (Borgi et al., 2017).

Hand in hand with this technology, artificial intelligence (AI) is used, which is based on a set of machines that, through defined algorithms, develop tasks that would normally require the intelligence of a person (Jakhar and Kaur, 2020). In addition, AI techniques such as machine learning (ML) are applied, which is defined as computational procedures that with the help of input values allow a machine to simulate human intelligence by adapting and learning from its environment without being explicitly programmed to do so (El Naga and Murphy, 2015). Within this branch there are different learning models, including normality models that are trained only with normal data without failures, i.e. healthy data, which represents a relevant advantage over a supervised model, which is trained with a history of previously labeled data between healthy or damaged and thus learns to predict the output value (Kotsiantis et al., 2006).

Processing and analysis of the data allows for a better understanding of the behavior and dynamics of the system. Predictive maintenance allows you to take advantage of this information to intervene only when the machine really needs it, providing advantages such as reduced maintenance costs, emergency stops,

inventory, and an increase in the useful life of spare parts and production efficiency, among others. (Peres et al., 2018; Sezer et al., 2018).

50 Table 1 shows scientific investigations based on the detection of failures classified according to the principal component or system of the WT, such as the gearbox, the pitch system, the yaw system, among others. Similarly, Table 2 shows the research based on fault detection but classified according to the monitoring method used.

**Table 1.** Papers based on the detection of faults classified according to the main component or system of the WT.

| | |
|---|---|
| 1. Gearbox | (Jiang et al., 2019; Zhang et al., 2012; Teng et al., 2016; Feng et al., 2014) (Dameshghi et al., 2019) |
| 2. Electrical and electronics | (Kim et al., 2011; Borchersen and Kinnaert, 2016; Astolfi et al., 2019) |
| 3. Blades/pitch angle | (Rezamand et al., 2020; Guo et al., 2021; Oliveira et al., 2020; Ou et al., 2017) (Dervilis et al., 2014) |
| 4. Tower | (Vidal et al., 2020; Nguyen et al., 2015, 2018; Motlagh et al., 2021) |
| 5. Bearing | (Natili et al., 2021; Encalada-Dávila et al., 2021; Hu et al., 2018; Fuentes et al., 2020) (Zhang et al., 2022) |
| 6. Yaw system | (Chen et al., 2020) |

**Table 2.** Papers of investigations based on fault detection but classified according to the monitoring method used

| | |
|---|---|
| 1. Temperature (oil, generator, bearing) | (Fu et al., 2019; Li et al., 2021; Zeng et al., 2020) (Guo et al., 2018) |
| 2. SCADA data | (Xiang et al., 2021; Chacón and Márquez, 2021; Wang et al., 2019) (Encalada-Dávila et al., 2022; Tutivén et al., 2022) (Dao, P. B., 2022; Dameshghi et al., 2019) |
| 3. Vibration signals | (Son et al., 2018; Ren et al., 2019; Joshuva et al., 2020) (Xiuli et al., 2018) |
| 4. Electrical signals | (Dahiya, R., 2018; Zhang et al., 2019) |
| 5. Acoustic emission | (Karabacak and Özmen, 2022; Yao et al., 2021) |
| 6. Strain measurements | (Hubbard et al., 2021) |
| 7. Other non-destructive testing | (Wang et al., 2022) |

For WTs, one of the predominant systems is the supervisory control and data acquisition (SCADA) and condition monitoring (CM) system (Artiago et al., 2018; Leite et al., 2018). CM systems are born from the need to guarantee a more controlled and safe operation. Most of these systems available on the market for WTs are mainly based on vibration measurements or in some works, such as Wang et al. (2019); Hammed et al. (2009), external variables and tower oscillations are also analyzed. This system allows moving from preventive maintenance to predictive maintenance, which, based on the study of behavior, allows a deviation to be detected, warning of a possible failure in the machine. However, its use requires the installation of additional sensors, which allow the acquisition and analysis of new variables, which causes an increase in the costs of operation, maintenance, and complexity. Regarding SCADA data, it has approximately 200 stored variables, which are acquired with a sampling frequency of 1 Hz, but, in general, to optimize space and bandwidth, the variables are only recorded every 10 minutes, storing the mean, standard deviation, minimum, and maximum values of each variable. This information, added to the work order records, results in the generation of complete operation histories, faults, and alarms for each WT.

Recently, there has been lot of interest in the analysis of SCADA data because although the system was not developed for condition monitoring, the extraction of relevant information through data analysis and processing can become a great alternative that allows one to ensure reliability, maintaining the costs and complexity of the machine, in this case WT. However, it is necessary to consider relevant points when using this information, such as the fact that there is great variability in the data due to changes in environmental conditions such as wind speed and direction, ambient temperature, turbulence, etc. Also, as explained above, in most cases the data is stored only in 10-minute intervals, so there is a loss of information. At the same time, work order history and entry are not always standardized and are prone to human error. Some researchers have already started to use SCADA data for fault detection in WTs using different tools and methodologies. As, for example, in Dao et al. (2018), methodologies based on SCADA data cointegration analysis are presented for condition monitoring and fault diagnosis, using data obtained from a single WT with a nominal power of 2 MW under environmental and operating conditions. In Xiang et al. (2021) an early warning model for fault detection is proposed, using a convolutional neural network (CNN) and an attention mechanism (AM) based on a long-short-term memory (LSTM) neural network, where SCADA data are used as input variables and build the CNN and LSTM architecture, and AM is applied to strengthen the impact of important information. Furthermore, with respect to studies on bearing failures in WT, in Zhang and Wang (2014), a bearing failure detection method based on existing SCADA data is proposed using an artificial neural network (ANN). Similarly, in Encalada-Dávila et al. (2022), a methodology is shown that detects main bearing failures using a gated recurrent unit (GRU) neural network.

Early detection of main bearing failures of wind turbines is crucial to guarantee the reliability of the element, as well as a safer and more efficient operation in wind farms. The main bearing is one of the most critical components in a WT, and a failure in it can cause significant damage to other components, such as the gearbox, generator, and blades, and result in downtime and expensive repairs, see Carrol et al. (2016). Early detection of main bearing failures enables predictive maintenance, giving maintenance crews time to plan and schedule repairs during low wind periods, minimizing the impact on energy production. Bearing damage in wind turbines can occur in different locations, including the rings, raceways, rollers, and cage. The most common types of bearing damage are related to heat release, which can result from friction, wear, and cracks, see Harris et al. (2006). All of these damage modes can significantly impact the lifetime of the bearing, which in turn can cause significant downtime and maintenance costs. Early detection of bearing damage through monitoring and detection of heat release can allow for timely repairs and maintenance, minimizing the impact on the bearing and other components, and reducing downtime and maintenance costs. The methodology proposed in this work aims to detect heat release in the bearings, allowing for early detection and diagnosis of potential bearing damage.

Although the topic of fault detection in main bearings of wind turbines has been the focus of numerous studies, as can be seen from the aforementioned references, in this paper a novel approach to this problem is presented based on principal component analysis (PCA) and data mining of only SCADA data. It should be emphasized that the stated methodology relies only on exogenous variables (ambient temperature and wind speed) and the temperature of the main bearing (internal variable most related to the target component, the main bearing), facilitating to isolate the faults that influence that one internal variable. In addition, all variables used in the strategy are readily available in all industrial-size wind farms (both older and newer), making it a practical and cost-effective solution for early fault detection. Cost is a critical factor in the renewable energy industry, and wind turbines are no exception. While advanced sensors and machine learning methods can provide more accurate and comprehensive data on wind turbine health, they also come with a higher price tag. In contrast, the proposed approach aims to offer a more affordable solution that can be easily adopted by wind farms that lack condition monitoring systems. This approach may be particularly beneficial for older wind turbines that lack the built-in sensors and monitoring capabilities of newer models. By extending the operation of wind turbines close to their expected service lifetime, the proposed approach can help wind farms generate more electricity and revenue over time. This not only improves the profitability of the wind park, but also increases the overall efficiency of the renewable energy sector. The longer a wind turbine operates, the more energy it generates, and the more emissions it can help offset. Furthermore, the proposed approach could help reduce the environmental impact of the renewable energy industry. Manu-

facturing new wind turbines requires significant amounts of energy and resources, so extending the life of existing turbines can help to reduce the need for additional production, promoting a more sustainable and

circular economy for wind energy.

The organization of this article is as follows. The general description of the wind park is described in Section 2. Section 3 presents the acquisition and analysis of the SCADA data used. In Section 4, the pre-processing of the data (data cleaning and variable selection) is presented. Then, in Section 5, the anomaly detection algorithm is shown. In Section 6 the indicators are analyzed, and the threshold value is defined.

The results are presented and discussed in Section 7 and finally, in Section 8, the conclusions are detailed.

## 2  Wind park description

In this study, data are collected from a wind park composed of 18 WTs located in Poland. All WTs have the same characteristics, which are detailed in Table 3. Figure 1 shows the principal components of these WTs.

**Table 3.** The main technological features of the type of WT in the wind park considered.

| | |
|---|---|
| Number of blades | 3 |
| Nominal power | 2300 kW |
| Voltage | 690 V |
| Rotor diameter | 101 m |
| Wind class | IEC IIb |
| Swept area | 8000 m$^2$ |
| Rotor speed | 6-16 rpm |
| Tower height | 80 m |
| Cut-in wind speed | 3-4 m/s |
| Rated wind speed | 12-13 m/s |
| Cut-out wind speed | 25 m/s |
| Gearbox type | 3-stage planetary/helical |
| Gearbox ratio | 1:91 |
| Power regulation | pitch regulation with variable speed |

Technical details of the wind turbines under study are out of the scope for the analysis presented in this

paper. However, it should be noted that wind turbine design and operation can impact the performance of fault detection methods. The book of Hansen  (2015), on the aerodynamics of wind turbines, provides a

comprehensive overview of wind turbine design and operation, including factors that can impact the accuracy of fault detection methods. Therefore, we encourage readers who are interested in the technical details of wind turbine design to refer to this resource.

Regarding the drivetrain configuration, three-point and four-point suspensions, which refer to one or two main bearings, respectively, are the most common wind turbine drivetrain architectures. In the three-point suspension configuration, which is the one used in the wind farm under study, the rotor is rigidly connected to the main shaft, which is supported by a single main bearing near the rotor. A shrink disk typically connects the downwind side of the shaft to the low-speed stage of the gearbox. The gearbox is supported by two

torque arms that are connected to the bedplate elastically. These two torque arms, along with the single main bearing, provide a total of three points of support. Furthermore, there are different types of state-of-the-art main bearings, as fully explained in Wenske (2022). In particular, the turbines of this park are equipped with the so-called spherical roller bearing (SRB) type. SRBs are characterized by their outer raceways being a portion of a sphere. The rollers, in turn, are shaped so that they closely conform to the inner and outer

raceways. This results in a bearing that is internally self-aligning and has a high radial load carrying capacity, please see Hart et al. (2019) for a more detailed explanation.

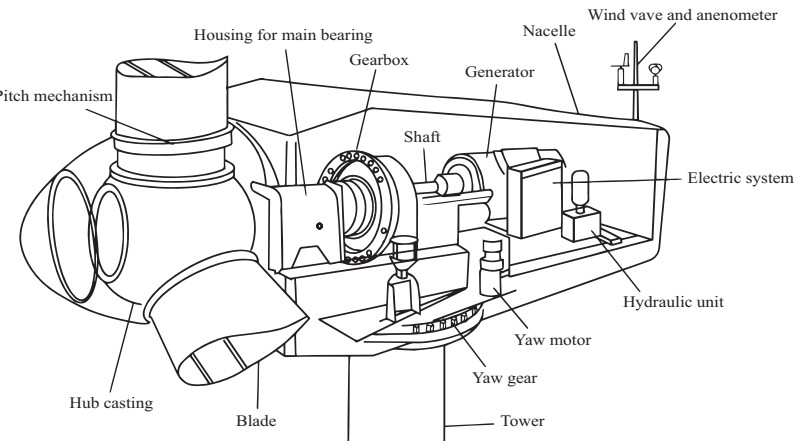

**Figure 1.** Main components of WTs. (Jiang et al., 2017).

Finally, each turbine is equipped with a SCADA system, thus plant operators can get an overview of the entire wind park. The main features of the SCADA system are the following: use of the standard communication protocol OPC UA, standardized data structure according to IEC61400-25, alarm and data history,

 online/offline trends. These SCADA-collected data are used in this work and are further explained in Section 3.

## 3 Data description, acquisition, and analysis

Wind Park SCADA data were recorded every 10 minutes from 1 January 2014 to 12 December 2019, obtaining the average, standard deviation, highest value, and lowest value of the data sampled from each variable.

These variables can be divided into different groups: energy, hydraulic, climatic, control, and heat emanating from the elements (see Encalada-Dávila et al. (2021)).

For the analysis and development of this work, it was decided to use only the mean of the external SCADA measurements (which can be broadly grouped into the so-called environmental variables; see Table 4) and the mean of the variable MainBTmp (low speed shaft temperature), related to our failure of interest, with

the objective that detection alerts are not affected by other types of failure other than the one under study. It is important to know that there is additional information (work orders) such as the maintenance and repair work carried out on each of the turbines, the dates on which they were carried out, and different references that help determine which elements were replaced or repaired. This information allows for the validation of the proposed methodology.

**Table 4.** External selected measurements.

| Variable | Description |
|----------|-------------|
| AmbieTmp | Ambient temperature |
| A1ExtTmp | Ambient temperature (different sensor) |
| PrWindSp | Primary wind speed |
| SeWindSp | Secondary wind speed |
| AcWindSp | Actual wind speed |
| PriAnemo | Primary anemometer reading |
| SecAnemo | Secondary anemometer reading |

### 165 3.1 Data division

To avoid biases in the normality model to be developed, before analyzing the data and performing the data preprocessing, it is important to perform the data division.

The data division procedure is a crucial step in any deep learning model, as it has a very relevant effect on the whole process. To separate the data for this study, the training data is selected such that:

– The model is trained using only healthy data.

    – The model is trained using data from all operating regions of the WT. There is no selection of a specific region or conditions of operation, or specific year season, thus the model is robust to all these variations.

As a result, a normality model is constructed that is tolerant to the various environmental and working
conditions of WTs.

Based on the work orders, for this analysis, the selection of WT number 5 of the wind park (from now on denoted as "WT5") is made, since it is known that in 2018 it suffered a failure of the main bearing, which is the failure of interest in this work.

For data division, it is taken into account that the main bearing failure occurs on 11 June 2018, and the
model must be trained for at least one year to ensure that it is seasonally robust. Therefore, according to the conditions detailed above, the data used to train the model are selected from 1 January 2014 to 11 December 2017, while the test data from 11 December 2017 to 12 December 2019, as shown in Figure 2.

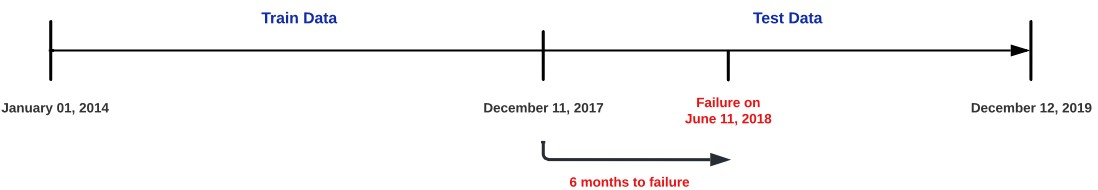

**Figure 2.** Data splitting for model training and testing.

It is acknowledged that the availability of almost three years of data may not always be feasible. However, a data length of this magnitude was deemed necessary to fully capture the normal operating behavior of the
main bearing and to establish a reliable baseline for fault detection. It was observed that when using one year of training data, the results were similar, but when the training data was reduced to only six months, the method was incapable of learning a normality model robust to all wind turbine operating scenarios, see Turnbull et al. (2022). Therefore, for the proposed approach, a minimum of one year of data is strongly recommended, and the methodology will significantly benefit from two or three years of available data.

Before performing the pre-processing of the data, the training data is explored to visualize the behavior of the different variables, as detailed in Section 3.2.

## 3.2  Exploratory data analysis

Exploratory data analysis (EDA) is a process of analyzing and understanding the input information, its structure, selecting relevant variables, cleaning outliers, etc. (Vanawat et al., 2021). That is, it seeks to explore

the data and discover important information through statistical graphs or some other visualization method, ensuring the quality and efficiency of the subsequent analysis (Shin, 2020).

It should be noted that, in general, failure to perform an EDA on the study data set results in the development of inaccurate models due to data bias, which are missing or inconsistent values. Therefore, in this study, the EDA is based on understanding the behavior of each of the variables relevant to the study, taking

into account that only training data are used. Figure 3 details the behavior of the eight selected variables, and the following deductions are obtained:

–   The plot of the MainBTmp variable shows seasonal variation as well as A1ExtTmp and AmbieTmp, indicating that the low-speed shaft temperature is altered by the environment temperature.

–   The graphs of the SeWindSp and SecAnemo variables show a very similar behavior. The same is

observed for the variables AcWindSp and PrWindSp.

–   In general, all variables show an increase in variations in wind speed at the same stages, so the measurements could be affected by wind currents.

–   For the PriAnemo variable, it can be observed in its behavior that it is a staggered response, where from 2014 to the beginning of 2018 it remained at the value of 1.2 while the remainder at almost 0.

This variable can be interpreted as an on- and off state, so it is decided to eliminate this variable, as it does not provide relevant information.

To correct for missing values, blanks, or outliers, pre-processing techniques must be used. For this reason, data pre-processing is proposed, which allows solving the problems mentioned above by converting them into quality data.

## 4  Data pre-processing

When working with SCADA data, there is the possibility that incomplete, inconsistent, unprocessed, or error-prone values may be present, so pre-processing is of great importance to ensure data quality through

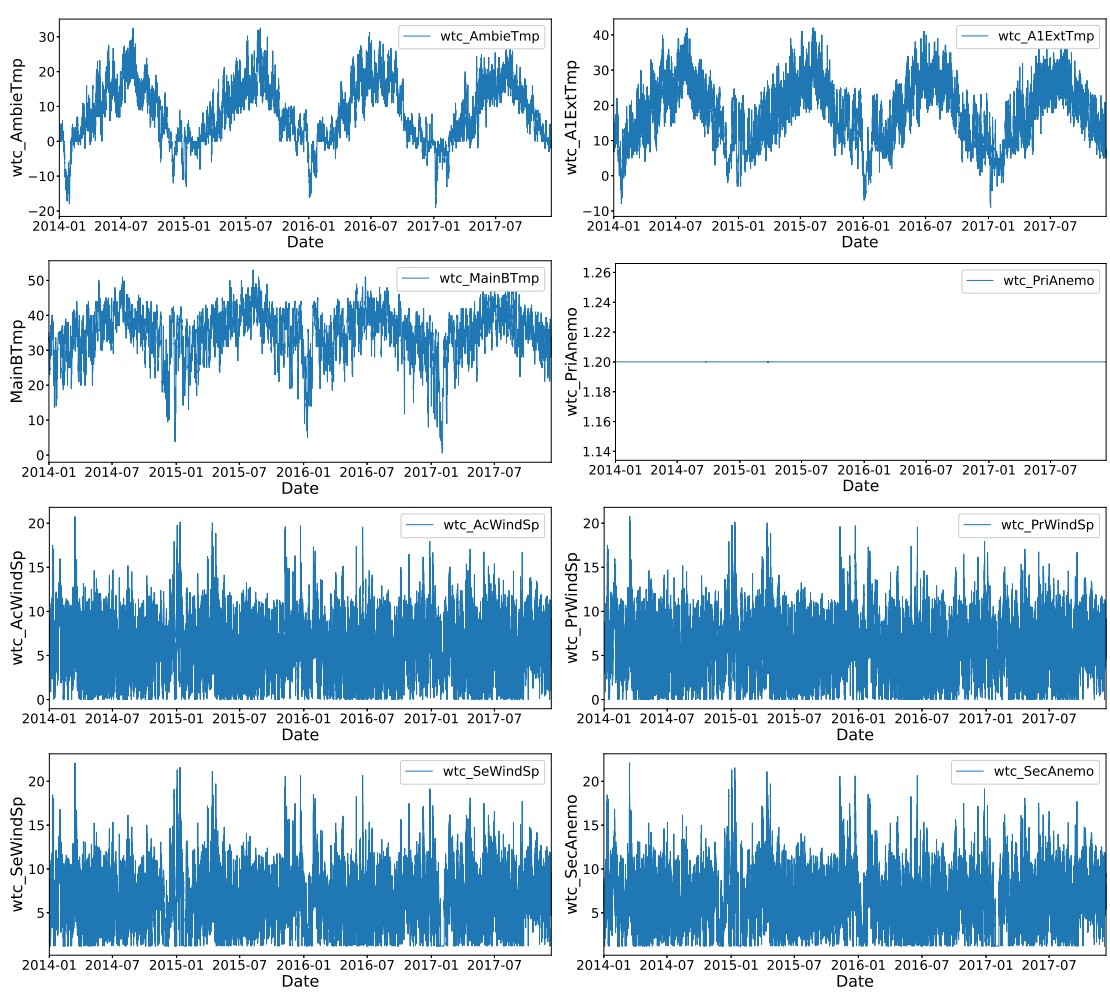

**Figure 3.** Exploratory data analysis for the selected variables.

data cleaning and preparation (García and Luengo, 2015). In this study, data cleaning and variable selection preprocessing techniques are used and explained in detail in the next subsections.

## 4.1  Data cleaning

Data cleaning is based on the determination and correction of missing or atypical values, where data are treated to give coherence to the data set (Brownlee, 2020). Several forms of outlier identification are available. In our study, extreme values (outliers) were not systematically removed since doing so could lead to a loss of information related to fault detection, as stated in Encalada-Dávila et al. (2021). Instead, a strategy of defining ranges based on realistic values that can be obtained by different sensors was adopted. This approach, which allows potentially useful information to be retained while still addressing the issue of outliers, was chosen. To ensure appropriate definition of the ranges, non-restrictive criteria were used that were wide enough to encompass the majority of the observed data. By adopting this approach, it is almost ensured that the only outliers removed are those related to non-working sensors (not well calibrated or with faults) and/or due to problems with the communication of the data, rather than outliers related to the underlying physical process being monitored. Table 5 shows the realistic possible values for each of the selected variables. All detected outliers are converted to missing values and subsequently filled with realistic values, preventing data waste and without introducing errors into the model.

**Table 5.** Range of possible values for each of the variables under analysis.

| Variable | Range |
|---|---|
| AmbieTmp | [-19    43] ℃ |
| A1ExtTmp | [-19    43] ℃ |
| MainBTmp | [0    100] ℃ |
| SeWindSp | [0    23] m/s |
| AcWindSp | [0    23] m/s |
| PrWindSp | [0    23] m/s |
| PriAnemo | [0    24] m/s |
| SecAnemo | [0    24] m/s |

The method described in Encalada-Dávila et al. (2021) is used for the imputation. To complete the missing data for two samples, the piecewise cubic Hermite polynomial interpolation method is used, which allows

new values to be included, ensuring that the function will behave monotonically and that the first derivative will be continuous. Also, since there may be missing values at the boundaries, the closest data that follows or precedes the missing values is used.

An example of the imputation strategy applied to complete the absent values in the low-speed shaft temperature is detailed in Figure 4.

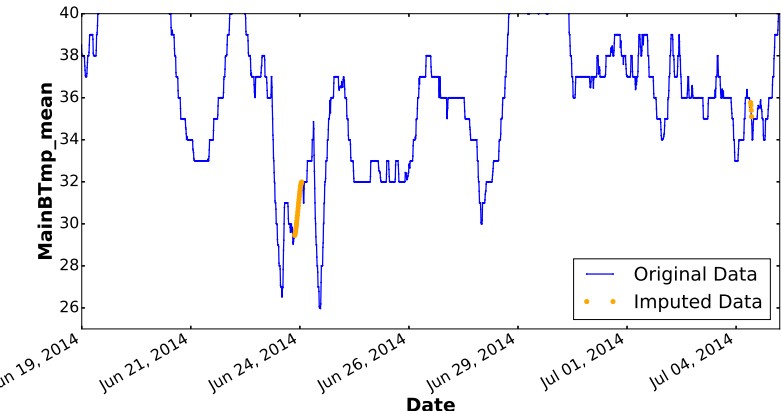

**Figure 4.** Imputation strategy used to fill missing values in the low-speed shaft temperature (given in Celsius degrees).


### 4.2 Variable selection

Recall that in Section 3 a preselection of variables is carried out based on the use of exogenous variables and the variable most related to the failure under study, but when working with large data sets (in this case several years), a new process of selecting variables is needed to reduce the number of features, eliminating 245 non-informative or redundant variables that may slow down the development and training of the model or also can introduce errors (Kim et al., 2011). Therefore, this procedure is carried out with the objective of improving the prediction efficiency of the model, providing faster and more reliable predictions.

One way to determine which variables are useful when training the model is through a correlation analysis, which allows one to know the relation between the variables. In this work, Pearson and Spearman are used. 250 Both Pearson and Spearman correlation coefficients are measures of the strength and direction of a linear relationship between two variables. The Pearson correlation coefficient is used when both variables are continuous and have a linear relationship. It measures the degree to which two variables are linearly related, and ranges from -1 (perfect negative correlation) to 1 (perfect positive correlation), with 0 indicating no correlation. The Pearson correlation coefficient assumes that the data are normally distributed. On the other

hand, the Spearman correlation coefficient is preferred when the variables are not normally distributed or are ordinal (ranked). It measures the degree to which two variables are monotonically related, meaning that they move in the same direction but not necessarily at a constant rate. Like the Pearson coefficient, Spearman correlation coefficient ranges from -1 to 1, with 0 indicating that there is no correlation.

According to Sabilla et al. (2019) depending on the coefficient value, it can be determined whether the
correlation between two variables is strong or weak, considering a strong relationship for values from 0.50 to 0.69, a very strong relationship from 0.70 to 0.89 and almost perfect from 0.90 to 1.

For the selection of the variables that are to be analyzed and studied, in this work it has been decided to select variables with a correlation greater than $+0.8$ and less than $-0.8$, selecting one of them and eliminating the other, given that, as observed when analyzing the data, there are variables that offer the same information
as others since they are highly correlated. Therefore, the selected variables are MainBTmp, SecAnemo and AmbieTmp, already described in Table 4. Figure 5 shows the correlation analysis performed between all variables.

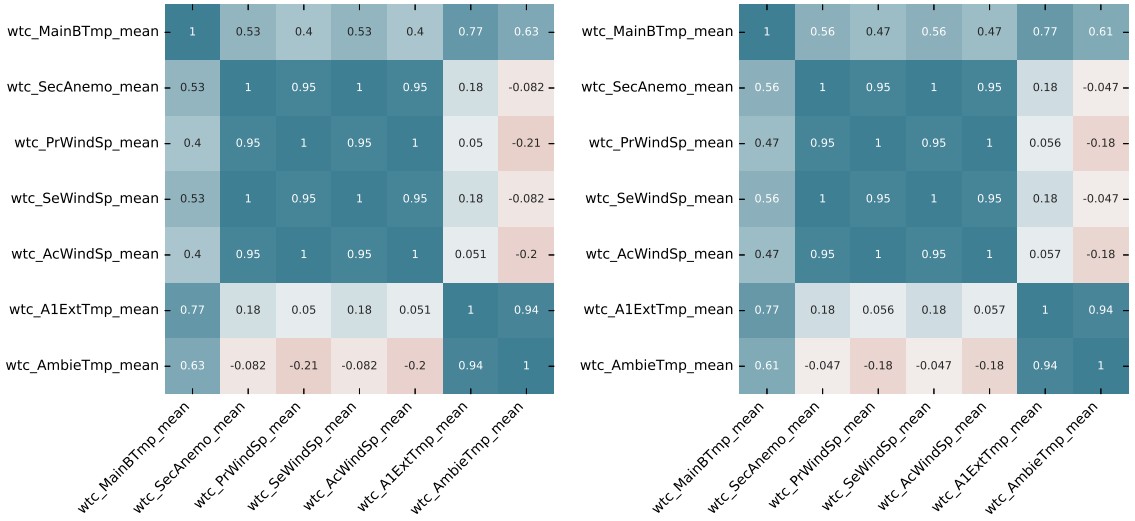

**Figure 5.** Pearson correlation heat map of preselected variables (left), and Spearman correlation heat map of preselected variables (right).

It is important to consider that to avoid errors due to seasonality, the temperature of the low speed shaft (MainBTmp) is subtracted from the environmental temperature (AmbieTmp). As you can see in Section 5,
the model is trained with this new processed variable and the SecAnemo variable.

## 5 Principal Component Analysis (PCA)

Principal component analysis (PCA) is a statistical technique widely used for the study of data in almost all scientific areas (Kurita, 2019). This method is based on determining the most significant base of a given data set to decrease its dimensionality for large data sets, simplifying the complexity of multidimensional spaces while keeping as much information as possible (Rodrigo, 2020). Its objective is to be able to differentiate which features are the most relevant, which are redundant and which are just noise (Tibaduiza et al., 2011).

Suppose that there is a sample of *n* observations with dependent variables that are interrelated and have noise (Jolliffe and Cadima et al., 2016), where each has *p* variables or dimensions. PCA is expected to obtain hidden and meaningful data from these observations and filter out the noise. To achieve this purpose, PCA determines a set of new orthogonal variables called principal components, *z*, which are the result of linear combinations of the initial variables (Kurita, 2019).

In short, with PCA it is possible to describe an observation, for which *p* values were previously needed with only *z* values, where $z < p$, as shown in Figure 6.

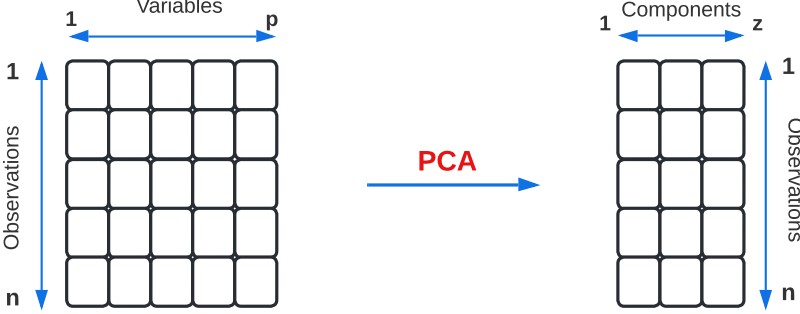

**Figure 6.** Dimensionality reduction applying PCA.

Suppose that there is a data set $X = [x_1, x_2, .., x_N]$, where each column is an observation *n* with *p* variables or dimensions. The mean vector of the sample $\bar{x}$ and the sample covariance matrix $\Sigma$ are defined in Equations (1) and (2), respectively, where the matrix $\widetilde{X} = [x_1 - \widetilde{x}, .., x_N - \widetilde{x}]$.

$$\bar{x} = \frac{1}{N} \sum_{i=1}^{N} x_i \tag{1}$$

$$\Sigma = \frac{1}{N} \sum_{i=1}^{N} (x_i - \bar{x})(x_i - \bar{x})^T = \frac{1}{N} \widetilde{X} \widetilde{X}^T \tag{2}$$

A principal component analysis stands to reason if the variables have strong correlations, as this is an indication of the presence of redundant information, and therefore few components could explain much of the total variance. The selection of the number of components is determined based on % of the variability that is considered appropriate. The first component collects the greatest possible variability of the original data, the second collects the maximum possible variability (in an orthogonal direction to the first one) not considered by the first, and the following components repeat it until an adequate variability is reached.

In (Kurita, 2019), the PCA technique is explained in greater depth, describing the mathematical formulas used to calculate each principal component and its projection in a new vector space.

One of the most important applications of PCA is its use in anomaly detection, where the normalized reconstruction error (RE) is used as a scoring level to determine whether a sample is healthy or anomalous. For this study, this technique is used, as described in Section 5.1.

## 5.1 Anomaly Detection with PCA

Anomaly detection with PCA is an unsupervised statistical strategy to identify anomalous samples present in a set of data elements, based on RE results (Rodrigo, 2020). In particular, the goal is to decompose the source data set into its principal components and then reconstruct the original data using the necessary principal components.

For this work, the preselected variables are projected using the PCA strategy, obtaining three principal components.

Equation (3) describes the first principal component, where $a_1^T = (a_{11}, .., a_{N1})^T$ is a set of weights for linear combinations.

$$y_{1i} = a_1^T (x_i - \bar{x}), (i = 1, .., N) \tag{3}$$

The vector of this component has a tendency in the direction of the observations with greater variance. The projection of each observation in that direction represents the value of the first component of that observation.

For the calculation of the following main components, the same principle is used iteratively until the original number of variables. Considering that the vector of each next principal component will represent the next direction in which the data have more variance and that it should not be correlated with the previous com-

ponent, which implies that the direction of their vectors is perpendicular/orthogonal to each other (Rodrigo, 2020).

Each new input is processed by the anomaly detector, where it first calculates the projection of the eigenvector and then the RE (Microsoft, 2020). The anomaly detection with PCA indicates that the anomalous elements will be the reconstructed data that differ the most from their original counterparts. See Figure 7.

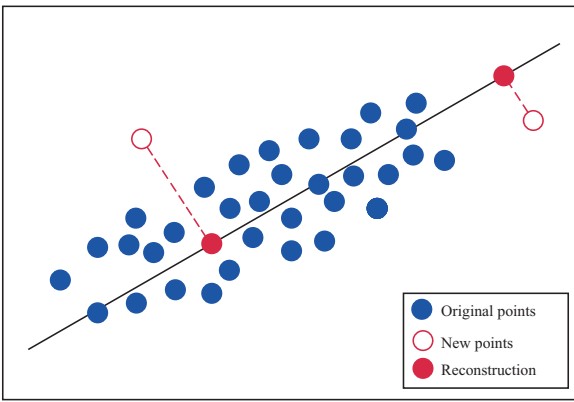

**Figure 7.** Anomaly detection employing PCA RE.

Let $x_n$, where $n = (1, 2, .., N)$ and $x_n \in \mathbb{R}^d$ be the set of data observations and $\widetilde{x}$ the reconstruction produced based on an implicit variable of smaller dimension. Then, the RE is defined by Eq. (4).

$$RE(x) = \frac{1}{N} \sum_{n=1}^{N} \| x_n - \bar{x}_n \|^2 \tag{4}$$

In this study, the PCA method is used with the objective of identifying observations that differ from a large part of the values, these being the healthy data used for the training process. When using this approach,
prefailure data (anomalies) are expected to have qualitatively distinct attributes from normal data. Therefore, the PCA anomaly detection model is trained only on data from one class (healthy samples), and subsequently predictions are made to find anomalies in the test data set.

Once the PCA anomaly detection algorithm has been created, all phases discussed above are represented in a diagram, as shown in Figure 8. Unlike other approaches, such as the neural autoencoder, the main utility
of using PCA as an anomaly detector is its simplicity. Figure 8 shows the data cleaning, variable selection, and training stages of the PCA anomaly detection model.

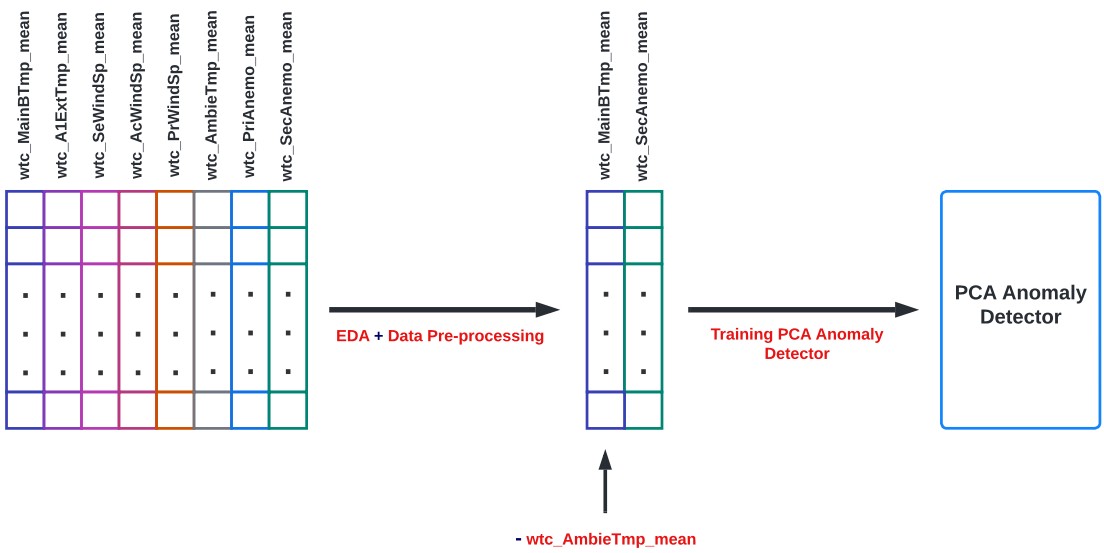

**Figure 8.** First phase of the stated methodology.

## 6 Fault detection indicator (FDI)

For FDI, a threshold is typically established. When the model's discovery of anomalies (outliers) exceeds the predetermined threshold, an alert is produced. However, given that the study only used available 10-minute
SCADA data, there may be an excessive number of false alarms, leading to alarm fatigue for WT operators. The FDI that is suggested to bypass this problem is computed in three phases, which are explained in this section. Weekly outliers are aggregated (counted) using the PCA anomaly detector approach, after which the exponential weighted moving average (EWMA) filter (Ratner, 2017) is applied to the weekly outlier count, and finally a threshold is defined.

**6.1 Weekly gathering**

SCADA samples for the current work are collected every 10 minutes, totaling 1008 samples per week. The anomaly detector with PCA algorithm's weekly grouping comprises counting the number of samples out of a total of 1008 that are labeled as anomalies. Formally, the number of samples identified as anomalies for the given $i$-th week is counted and indicated as $C_i$.

## 6.2   Exponential weighted moving average (EWMA)

A well-known technique for smoothing historical data is the moving average (MA) approach (Hunter, 1986). The MA has several expansions, each with its own unique features, but their fundamental goal is still the same. In this work, a common MA extension known as EWMA is applied, which combines the calculation of the weight factor for the weighted moving average (WMA) and exponential moving average (EMA). A WMA assigns each observation the same weight, but an EWMA responds more strongly to recent sample changes. A parameter, $\alpha$, must be defined to determine the importance of each sample in the EWMA calculation. This technique follows the starting time series more accurately, the higher the value of alpha. Weekly grouping time series, $C_i$, is applied to the EWMA formula used in this study as

$$\mathrm{EWMA}_t = \alpha C_t + (1 - \alpha)\mathrm{EWMA}_{t-1}, \tag{5}$$

where $\mathrm{EWMA}_0$ is the mean of previous data and $\alpha$ is the user-determined weight. One way to specify the parameter an is in terms of spans, generally known as an $n$-day EWMA,

$$\alpha = \frac{2}{s+1}. \tag{6}$$

In this work, $s = 4$, indicating that 4-week groups are taken into account (around a month). Actually, the conclusions of McKinnon et al. (2020) have an impact on this choice. They evaluated three different moving windows, daily, weekly, and monthly, in their study of the impact of time history on WT failures using SCADA data. The weekly moving window performs best in terms of discovering failures compared to the others.

The FDI reports alarms when the total number of anomalies exceeds a specified threshold. The EWMA filter and weekly counting are then used. The EWMA over the mean ($\mu$) and standard deviation ($\sigma$) of the training set is calculated. Recalling that values within three standard deviations of the mean make up roughly 99.7% of a data set with an approximately normally distributed distribution, the so-called three-sigma rule conveys a common heuristic that it is desirable to regard 99.7% of probabilities as being close to certain. As a result, the criterion used in this study specifies that results greater than three standard deviations from the mean should be regarded as anomalies. Consequently, the threshold is stated as

$$\text{threshold} = \mu + 3\sigma. \tag{7}$$

The flow chart shown in Figure 9 describes the latter phases of the approach (from the model of the PCA anomaly detector already trained). The graphic shows how the inferences of $X_{\mathrm{train}}$ are grouped by week

and then filtered to determine a threshold. Then the same procedure is used with the $X_{\text{test}}$ inferences, but this time the goal is to utilize the previously established threshold (from $X_{\text{train}}$) to trigger an alert when the output goes over it.

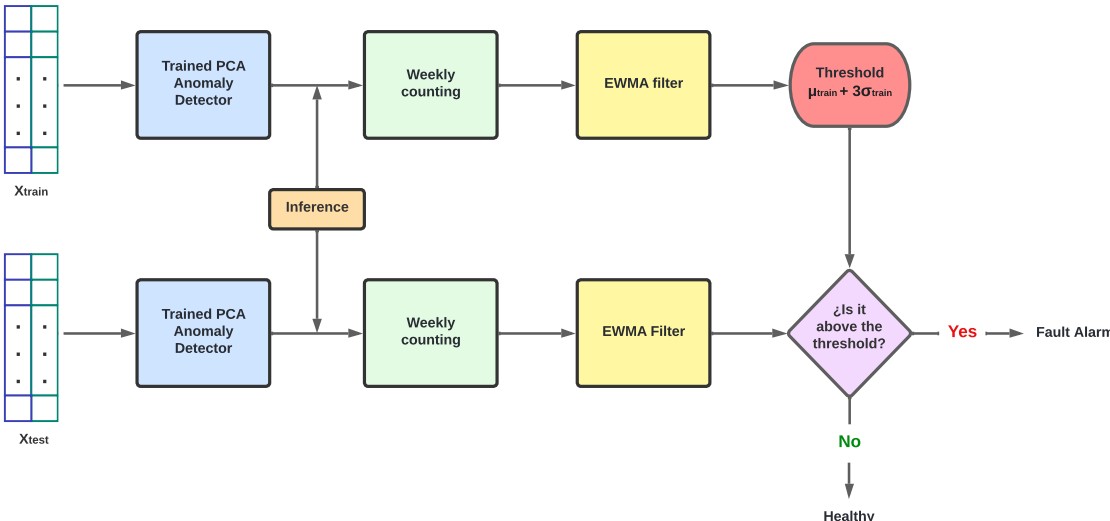

**Figure 9.** Second phase of the stated methodology.

## 7 Results

As a result of the proposed study, a predictive maintenance plan is developed to predict failures in the main bearing of the wind park. The model only needs healthy data to be trained and, as a result, allows for detecting anomalies before the fatal failure occurs.

Figures 10,11 and 12 show the figures of the weekly indicator, using the test data, for each of the 18 WTs, to which the detection threshold defined for the evaluation of the model is applied.

When analyzing the figure of each turbine, it can be seen that WT5 and WT16 have data above the established threshold, indicating the presence of alarms and predicts a failure during the indicated time.

In the case of the WT5 turbine, based on the information provided by the wind park, a work order was generated on 11 June 2018 for immediate maintenance of the WT5 turbine, corresponding to a main bearing failure (failure of interest in this study). Using the model, it can be seen in Figure 10 that 3 alerts were generated. The first alert generated is given on 9 April 2018, the next on 16 April 2018 and the last on the

day of failure (10 June 2018). This allows inferring that the system is capable of alerting about this type of failure months in advance, which allows corrective measures to be taken in time, to avoid emergency shutdowns, additional costs, and loss of efficiency in energy production. Note that after peaks (Figure 11, WT5), the signal drops sharply again for a long period. This is because the heat created from an initial failure mode (heating from an initial crack, friction, wear,...) is detected by the methodology, but its appearance is not continuous over time until the final breakdown. In contrast, when the failure mode advances, for example, when a crack propagates, the generated heat appears. When the crack remains still, no further heat is generated; thus, the alarm is set off. However, cracks are already present and can advance at any time, leading to the possible failure of the component. Thus, in this methodology, whenever the alarm is on (even when it is set off after a few weeks), it is highly recommended to check the specific WT.

It is significant that the proposed approach is designed specifically to detect (using only standard SCADA data, which are usually 10-minute averaged) the possible heat generated from an initial failure mode, such as the initiation or propagation of the crack, friction, electrical discharge and other failure modes associated with heat release. These types of failure typically result in a gradual and sustained increase in temperature (while they evolve), rather than sudden spikes or drops, which makes them detectable even with low sampling rates, as temperature variables have a low dynamic and still contain the information of the fault after being 10-minute averaged. With respect to the use of the weekly average, it is intended to reduce false positives by smoothing out transient fluctuations in the data that are not indicative of actual anomalies. Although this averaging may limit the resolution of the approach, as it could smooth out subtle changes in the data that could be indicative of early-stage anomalies, this trade-off is necessary to minimize false alarms and ensure practical utility of the methodology (and avoid alarm fatigue).

On the other hand, analyzing the case of the WT16 turbine, the model generated the alert on December 17, 2018. And on June 26, 2019, the wind park registered a work order in this WT, due to the presence of a fault in the gearbox. Although this is not the failure of the study, the detection of this type of failure may be caused by the high relationship between the main components where failures were recorded (low speed shaft and gearbox) and the use of variables or characteristics of the model that are mostly exogenous.

In summary, 18 wind turbines were examined, of which 16 were considered healthy and correctly classified as such. One turbine had the fault of interest and was correctly classified as faulty. Another turbine had a fault (that was not the fault of interest) and was classified as faulty, which could be considered a false alarm. However, in practice, the fact that an alarm was raised for a fault in a different component could still be useful, as it indicates the need for maintenance or further inspection. Therefore, in addition to the hit rate and error rate, the practical implications of false alarms should also be taken into account.

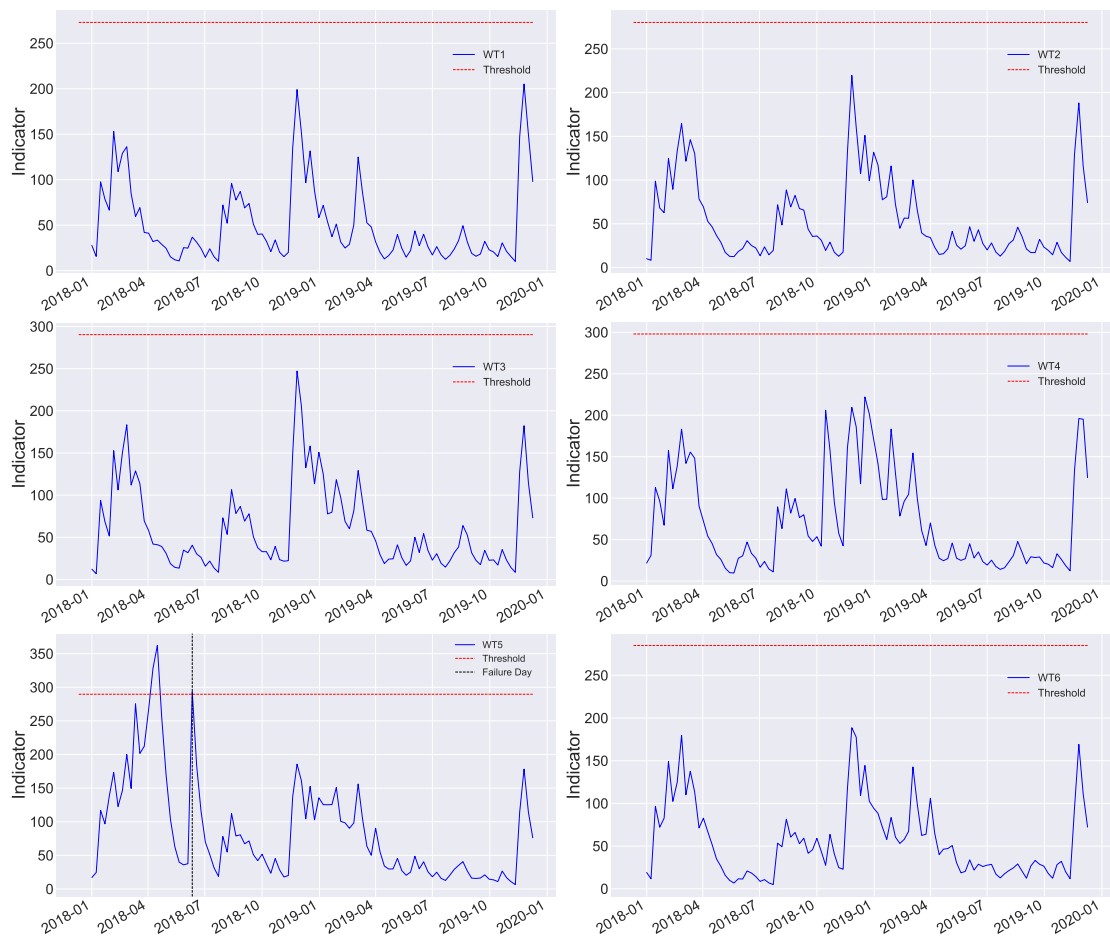

**Figure 10.** Weekly indicator (WT1-WT6).

## 8 Conclusions

In this work, an anomaly detection system is developed and evaluated by principal component analysis (PCA), in order to obtain early warnings of possible failures in the main bearing of the WTs. The stated methodology only requires healthy data at the time of development of the model, and the achievements obtained show a functional and efficient system in the generation of alerts for the detection of failures. Among the main advantages of this system are:

- It is not necessary to implement or purchase new sensors or additional systems, since only the data acquired by the SCADA system is used.

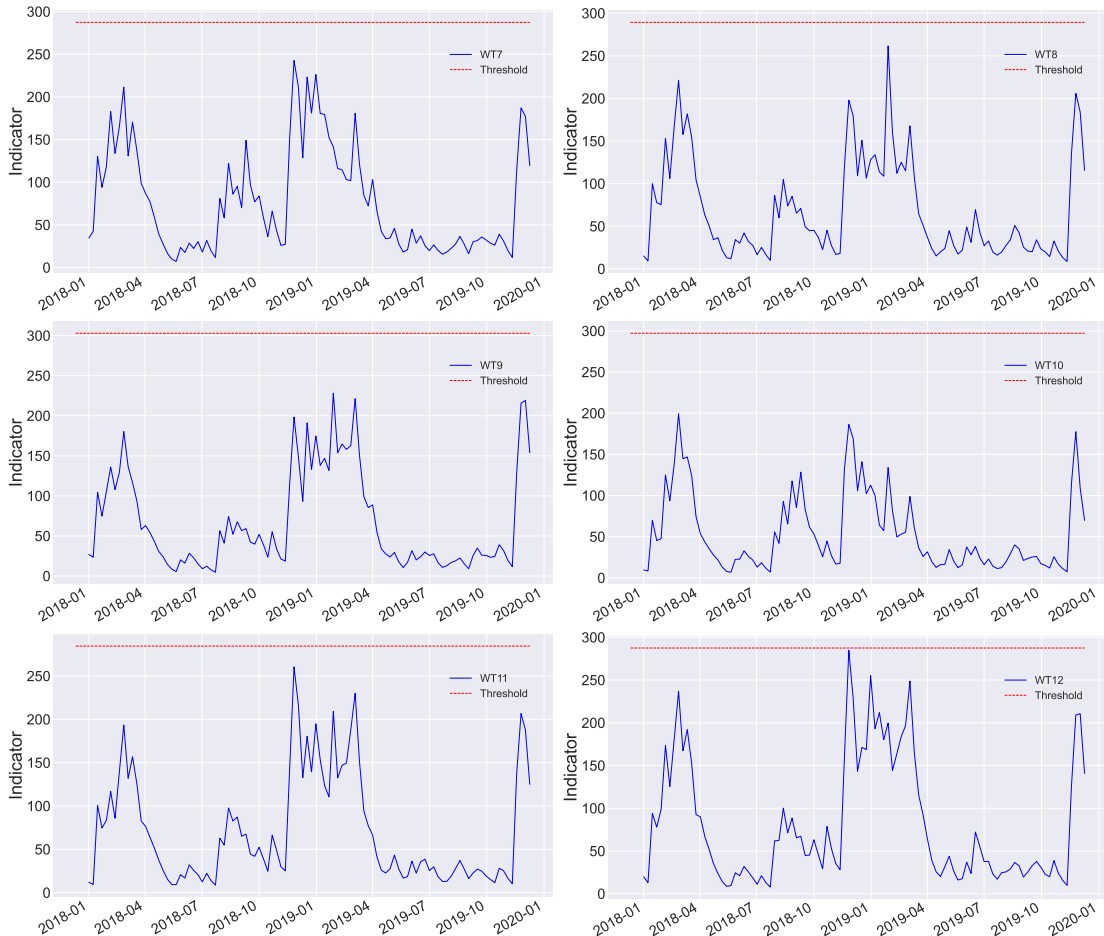

**Figure 11.** Weekly indicator (WT7-WT12).

– It is not necessary to have a turbine failure history, as this strategy is developed with a semi-supervised model that is developed with healthy information only.

– For the training and evaluation of the model, data from a real wind park in production are used.

It can be concluded that the stated methodology works accurately and efficiently since, when tested over a year in a complete wind park, it raised alerts months before the fatal breakdown occurs without false alarms. In addition, it allows for the detection of failures of not only the main bearing but also nearby components, thus offering the industry the benefit of predictive maintenance for early planning, guaranteeing long-term

safe and continuous operation, reducing downtime, emergency stops, and additional costs.

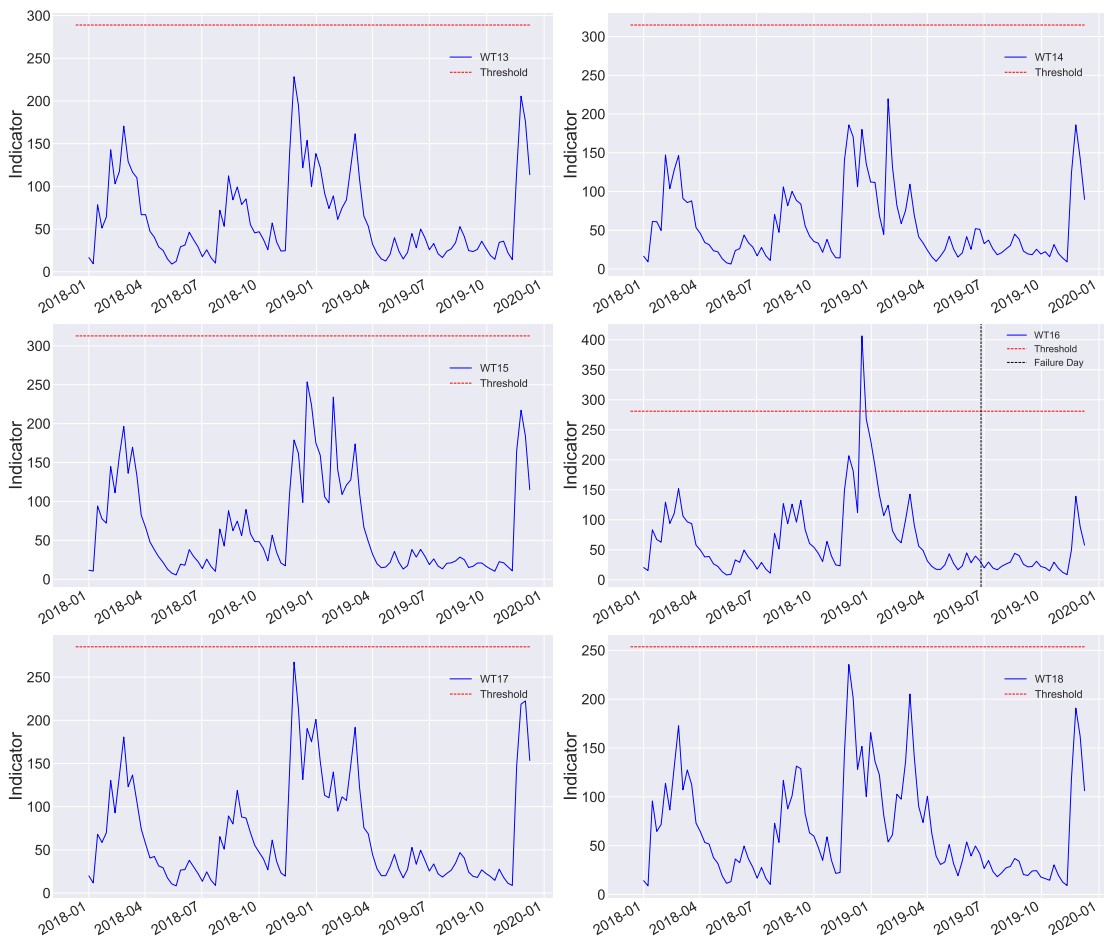

**Figure 12.** Weekly indicator (WT13-WT18).

Finally, while our approach has shown promising results, there are several areas for future research. One limitation of our approach is its applicability to new datasets with different characteristics, as each WT depends on its own model trained with its own data. In future work, we plan to explore the use of transfer learning to overcome this limitation and develop models that can generalize to new datasets. The results demonstrate that the stated approach is effective in detecting a main bearing fault that resulted in a significant increase in temperature. Although only one failure was available in the investigated wind park data, which is insufficient for statistical analysis, any bearing fault leading to heat release might be detectable by the proposed strategy. However, to more extensively investigate the performance of the model, it is necessary to apply it to other wind parks with main bearing failure issues. Therefore, future work will test the model


on a larger dataset to assess its performance in different scenarios and draw more generalizable conclusions. While the main bearing temperature was found to be a suitable indicator for detecting faults in wind turbines, as also stated in a recent paper by Murgia et al. (2023), another limitation of the proposed approach is that it cannot precisely locate the fault or its severity. Further developments could be pursued in this direction, for instance, by incorporating high-sampling data and/or additional sensors to improve the precision of the

fault location. However, this may come at the cost of increased complexity and expense, which is trying to be avoided in this work where the main objective is to contribute a cost-effective solution where all variables used are readily available in all industrial-size wind farms (both older and newer).

*Author contributions.* All authors have contributed equally.

*Competing interests.* No competing interests are present.

*Acknowledgements.* This work is partially funded by the Spanish Agencia Estatal de Investigación (AEI) - Ministerio de Economía, Industria y Competitividad (MINECO), and the Fondo Europeo de Desarrollo Regional (FEDER) through the research project PID2021-122132OB-C21. In addition, the authors express their gratitude and appreciation to the Smartive company (http://smartive.eu/), as this work would not have been possible without their support and ceded wind park data.

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
