# Peer review of "Anomaly-Based Fault Detection in Wind Turbine Main Bearings"

_Wind Energy Science, 2022_

## Referee Comment (RC2)

**Review**

**General comments**

In my opinion, the publication represents a useful contribution to scientific progress in the context of WES. It is of interest to the entire wind power community.

The main objective of the work is anomaly detection using simple Principal Component Analysis (PCA). The topic of artificial intelligence and machine learning is one of the hot topics of the moment. Therefore, it is also important to examine how these methods can lead to improvements in the context of wind turbines.

It is important to the authors to show a simple solution that does not require additional sensors. The approach using PCA and the SCADA data is interesting here. However, I wonder whether, given the value of the turbines and the maintenance costs that may be necessary, additional sensors and higher-quality ML methods would not be more effective. The state of the art is that PCA is not particularly suitable for anomaly detection.

**Specific comments**

Data preprocessing is not sufficiently described in the paper. If I understand it correctly, a range is specified for the real data and outliers are adjusted accordingly to the damage-free training data. This has several problems: Weak signals are filtered out, the model is only valid for the system under consideration, and the model assumes that the system under consideration is at the bottom of the bathtub curve, i.e. entirely error-free. Overall, due to the low sampling rate, the 10-minute intervals and the averaging over a week, the data appear to me to be very smoothed, which makes it difficult to find anomalies. Since we are dealing here with time series, a simple Pearson correlation is only of limited help (a Spearman's rank correlation should at least be examined here).

The data show a clear seasonal component. The question arises why this was not removed by decomposition, especially since the model is only based on individual data sets. After peaks (Figure 11), the signal drops sharply again for longer periods. What is the reason for this? Since the level seems to be significantly lower in the period from May to November, it is questionable whether incipient damage could be detected here at all.

Overall, despite the previously mentioned criticisms, the work is important because it helps to further advance the topic of machine learning and discuss the benefits and methods.

The anomalies also do not allow any statement on the type of damage present and the severity of the error. At the same time, there is no statement about the historical data and any false alarms. For a scientific consideration, a hit rate AND an error rate must be given.

The metrics in Table 4 need to be explained.

Line 133 and Figure 3: It is not described what kind of damage is typically involved here (lack of lubrication, wear, pitting, ...). Each defect should produce certain characteristics in the measured variables.

The structural, linguistic and graphic quality of the publication is very good. The work is clearly structured and the tables, graphs and pictures are easily recognizable and informative.

**Technical corrections**
Figure 16 with WT16 should be placed closer to line 326 where it is addressed.

---

## Author Comment (AC1)

**ANOMALY-BASED FAULT DETECTION IN WIND TURBINE MAIN BEARINGS**

Lorena CAMPOVERDE-VILELA, María del Cisne FEIJÓO, Yolanda VIDAL, José SAMPIETRO, and Christian TUTIVÉN

**Response to reviewers**

**General comments of the authors**

Dear Editor and the Reviewers,

We sincerely thank you for your constructive comments. Under the reviewers' comments and suggestions, the manuscript has been significantly strengthened both in contents and clarity. Below, you can see the changes that we made in response to each reviewer's comment.

The editor and reviewers found the paper of interest, yet they felt that several issues needed to be improved and clarified before the paper could be accepted for publication. In the revised manuscript:

- The changes made in response to Reviewer 1 are marked in blue.

- The changes made in response to Reviewer 2 are marked in red.

- The changes made in response to Reviewer 3 are marked in brown.

**Reviewer 1**

The manuscript entitled "Anomaly-Based Fault Detection in Wind Turbine Main Bearings" deals with a very interesting topic, which is perfectly adequate for the scientific objectives of the journal.

In a nutshell, the authors propose a PCA-based alarm raising method for diagnosing incoming damages to the main bearing of wind turbines. The method is based on SCADA data mining.

The work is well written and well presented. The workflow is very clear and presented in detail, such that it can be replicated by scholars.

The peculiarity of the work is that only exogenous variables (environmental) and the temperature of the component of interest (main bearing) are employed.

Therefore, in general I have a very positive opinion on this work. Nevertheless, there are some aspects which could be discussed more in deep.

**Author's reply**: Thank you for the positive feedback on the manuscript entitled "Anomaly-Based Fault Detection in Wind Turbine Main Bearings". We are grateful to hear that you consider the topic to be both interesting and well-suited to the scientific objectives of the journal. We also appreciate your comments regarding the clarity and replicability of the workflow, as well as the use of exogenous variables and temperature exclusively in the proposed method. Finally, we acknowledge your suggestions for discussing certain aspects of the work in greater detail. We will address them in this point-by-point answer to the suggestions given for improvement.

1. A considerable number of studies has been recently devoted to this topic. Therefore, I recommend that the authors highlight more clearly the innovative contribution and the points of strength of their work.

**Author's reply**: We appreciate your suggestion to highlight the innovative contribution and strengths of our work. In particular, the following paragraph has been added in the Introduction Section.

> Although the topic of fault detection in main bearings of wind turbines has been the focus of numerous studies, as can be seen from the aforementioned references, in this paper a novel approach to this problem is presented based on principal component analysis (PCA) and data mining of only SCADA data. It should be emphasized that the stated methodology relies only on exogenous variables (ambient temperature and wind speed) and the temperature of the main bearing (internal variable most related to the target component, the main bearing), facilitating to isolate the faults that influence that one internal variable. In addition, all variables used in the strategy are readily available in all industrial-size wind farms (both older and newer), making it a practical and cost-effective solution for early fault detection.

2. The authors employ almost three years of data for model training. For the necessities of real-time wind farm monitoring, it is not obvious that such amount of healthy data is available. Could the authors discuss their models' performance with shorter training data sets? I suggest the following reference: Turnbull, A., Carroll, J., & McDonald, A. (2022). A comparative analysis on the variability of temperature thresholds through time for wind turbine generators using normal behaviour modelling. Energies, 15(14), 5298

**Author's reply**: We appreciate your suggestion to examine the performance of our method with shorter training data sets. In response to this comment, the following paragraph has been added to Section 3.1 in the revised manuscript.

> It is acknowledged that the availability of almost three years of data

may not always be feasible. However, a data length of this magnitude was deemed necessary to fully capture the normal operating behavior of the main bearing and to establish a reliable baseline for fault detection. It was observed that when using one year of training data, the results were similar, but when the training data was reduced to only six months, the method was incapable of learning a normality model robust to all wind turbine operating scenarios, see Turnbull et al. (2022). Therefore, for the proposed approach, a minimum of one year of data is strongly recommended, and the methodology will significantly benefit from two or three years of available data.

Furthermore, the results for one year of training data, and only six months of training data are shown in Figures 1 and 2. These figures have not been incorporated into the manuscript, given that their inclusion would not suffice to provide a comprehensive analysis of the impact of distinct training periods. On the other hand, such analysis falls beyond the scope of the paper.

[Figure]

Figure 1: Results for WT1 to WT6 using one year of trainig data.

[Figure]

Figure 2: Results for WT1 to WT6 using six months of training data.

3. The authors obtain a result similar to that obtained, for example, in the recent paper Murgia, A., Verbeke, R., Tsiporkova, E., Terzi, L., & Astolfi, D. (2023). Discussion on the Suitability of SCADA-Based Condition Monitoring for Wind Turbine Fault Diagnosis through Temperature Data Analysis. Energies, 16(2), 620. The main bearing temperature is the most adequate target to monitor for raising an alarm, but there is an issue related to the capability of the model in locating adequately the fault. In this work, using the main bearing temperature, a fault regarding the main bearing itself and a fault regarding the gearbox are diagnosed similarly. This occurs also in the paper which I have indicated. Therefore, I am wondering if the authors have ideas for further developments regarding the issue of precise fault location.

**Author's reply**: Thank you for drawing our attention to the excellent recent paper "Discussion on the Suitability of SCADA-Based Condition Monitoring for Wind Turbine Fault Diagnosis through Temperature Data Analysis" by Murgia et al. In this work, as well as in our study, the main bearing temperature was found to be a suitable indicator for detecting faults in wind turbines. However, we acknowledge the issue you raised regarding the capability of the model in precisely locating the fault. As not being able to adequately locating the fault is a clear limitation of the proposed methodology, in the revised manuscript we added the following paragraph in the Conclusions Section together with a reference to Murgia et al. paper.

> While the main bearing temperature was found to be a suitable indicator for detecting faults in wind turbines, as also stated in a recent paper by Murgia et al. (2023), another limitation of the proposed approach is that it cannot precisely locate the fault or its severity. Further developments could be pursued in this direction, for instance, by incorporating high-sampling data and/or additional sensors to improve the precision of the fault location. However, this may come at the cost of increased complexity and expense, which is trying to be avoided in this work where the main objective is to contribute a cost-effective solution where all variables used are readily available in all industrial-size wind farms (both older and newer).

Finally, we would like to thank the reviewer for the valuable feedback and the time to review the paper.

---

## Author Comment (AC2)

**Anomaly-Based Fault Detection in Wind Turbine Main Bearings**

**Lorena Campoverde-Vilela, María del Cisne Feijóo, Yolanda Vidal, José Sampietro, and Christian Tutivén**

**Response to reviewers**

**General comments of the authors**

Dear Editor and the Reviewers,

We sincerely thank you for your constructive comments. Under the reviewers' comments and suggestions, the manuscript has been significantly strengthened both in contents and clarity. Below, you can see the changes that we made in response to each reviewer's comment.

The editor and reviewers found the paper of interest, yet they felt that several issues needed to be improved and clarified before the paper could be accepted for publication. In the revised manuscript:

- The changes made in response to Reviewer 1 are marked in blue.

- The changes made in response to Reviewer 2 are marked in red.

- The changes made in response to Reviewer 3 are marked in brown.

**Reviewer 2**

**General comments**

In my opinion, the publication represents a useful contribution to scientific progress in the context of WES. It is of interest to the entire wind power community. The main objective of the work is anomaly detection using simple Principal Component Analysis (PCA). The topic of artificial intelligence and machine learning is one of the hot topics of the moment. Therefore, it is also important to examine how these methods can lead to improvements in the context of wind turbines.

**Author's reply**: We appreciate your positive remarks on the contribution of the work to the scientific progress of wind energy systems and the importance of exploring the application of artificial intelligence and machine learning in the field of wind turbines.

It is important to the authors to show a simple solution that does not require additional sensors. The approach using PCA and the SCADA data is interesting here. However, I wonder whether, given the value of the turbines and the maintenance costs that may be necessary, additional sensors and higher-quality ML methods would not be more effective. The state of the art is that PCA is not particularly suitable for anomaly detection.

**Author's reply**: Thank you for this comment, for which we have improved the explanation of the real and practical utility of the proposed methodology in the revised manuscript. For this reason, we have added the following paragraph in the Introduction Section.

> Cost is a critical factor in the renewable energy industry, and wind turbines are no exception. While advanced sensors and machine learning methods can provide more accurate and comprehensive data on wind turbine health, they also come with a higher price tag. In contrast, the proposed approach aims to offer a more affordable solution that can be easily adopted by wind farms that lack condition monitoring systems. This approach may be particularly beneficial for older wind turbines that lack the built-in sensors and monitoring capabilities of newer models. By extending the operation of wind turbines close to their expected service lifetime, the proposed approach can help wind farms generate more electricity and revenue over time. This not only improves the profitability of the wind park, but also increases the overall efficiency of the renewable energy sector. The longer a wind turbine operates, the more energy it generates, and the more emissions it can help offset. Furthermore, the proposed approach could help reduce the environmental impact of the renewable energy industry. Manufacturing new wind turbines requires significant amounts of energy and resources, so extending the life of existing turbines can help to reduce the need for additional production, promoting a more sustainable and circular economy for wind energy.

Regarding the concern about the suitability of using Principal Component Analysis (PCA) for anomaly detection in wind turbines, we agree that this method may have some limitations, but it is well-suited for anomaly detection. In particular, PCA is a widely used technique for identifying patterns and trends in large data sets. By reducing the dimensionality of the data, PCA allows for the extraction of the most important information and the identification of the most significant factors contributing to the variance in the data. In the context of anomaly detection, this can help identify the most relevant features that contribute to the anomalous behavior. In our paper, the proposed approach using PCA (and SCADA data) has demonstrated promising results in detecting faults in the main bearings of wind turbines, as shown in the results obtained with real SCADA data.

**Specific comments**

Data preprocessing is not sufficiently described in the paper. If I understand it correctly, a range is specified for the real data and outliers are adjusted accordingly to the damage-free training data. This has several problems: Weak signals are filtered out, the model is only valid for the system under consideration, and the model assumes that the system under consideration is at the bottom of the bathtub curve, i.e. entirely error-free. Overall, due to the low sampling rate, the 10-minute intervals and the averaging over a week, the data appear to me to be very smoothed, which makes it difficult to find anomalies. Since we are dealing here with time series, a simple Pearson correlation is only of limited help (a Spearman's rank correlation should at least be examined here).

**Author's reply**: We apologize for the insufficient description of the data preprocessing in the initial submission. In the revised manuscript, we provide a more thorough and clear description of the data preprocessing. In particular, the following paragraph has been added, that also answers the reviewer comments about the problem of weak signals filtering out.

> In our study, extreme values (outliers) were not systematically removed since doing so could lead to a loss of information related to fault detection, as stated in Encalada et al. (2021). Instead, a strategy of defining ranges based on realistic values that can be obtained by different sensors was adopted. This approach, which allows potentially useful information to be retained while still addressing the issue of outliers, was chosen. To ensure appropriate definition of the ranges, non-restrictive criteria were used that were wide enough to encompass the majority of the observed data. By adopting this approach, it is almost ensured that the only outliers removed are those related to non-working sensors (not well calibrated or with faults) and/or due to problems with the communication of the data, rather than outliers related to the underlying physical process being monitored.

Thank you for raising the issue of the model's limited applicability beyond the specific system studied in our paper. The reviewer is correct that this is a potential limitation of the proposed approach. One way to address this limitation is through the use of transfer learning, which involves training a model on one dataset and then fine-tuning it on a new, related dataset. This can help to generalize the model to new datasets with different characteristics, and it is an area of active research in the field of machine learning. However, this is beyond the scope of this paper, as our goal was to develop and evaluate a model for each individual wind turbine based on its own data. We will consider exploring the use of transfer learning in future work, and this was added in the Conclusions Section with the following paragraph.

> Finally, while our approach has shown promising results, there are

several areas for future research. One limitation of our approach is its applicability to new datasets with different characteristics, as each WT depends on its own model trained with its own data. In future work, we plan to explore the use of transfer learning to overcome this limitation and develop models that can generalize to new datasets.

Regarding the comment about the correlation study, we agree that the Spearman's rank correlation is a useful tool for analyzing time series data, particularly when the relationship between variables may not be strictly linear. In response to the suggestion, we have re-examined our data using the Spearman's rank correlation, and we found that the results are consistent with our previous findings using the Pearson correlation. In the revised manuscript, the results obtained with the Spearman's rank correlation have been added, together with the addition of the following paragraph.

Both Pearson and Spearman correlation coefficients are measures of the strength and direction of a linear relationship between two variables. The Pearson correlation coefficient is used when both variables are continuous and have a linear relationship. It measures the degree to which two variables are linearly related, and ranges from -1 (perfect negative correlation) to 1 (perfect positive correlation), with 0 indicating no correlation. The Pearson correlation coefficient assumes that the data are normally distributed. On the other hand, the Spearman correlation coefficient is preferred when the variables are not normally distributed or are ordinal (ranked). It measures the degree to which two variables are monotonically related, meaning that they move in the same direction but not necessarily at a constant rate. Like the Pearson coefficient, Spearman correlation coefficient ranges from -1 to 1, with 0 indicating that there is no correlation.

Finally, in regard to the comment about the low sampling rate, we agree that this issue was not clearly stated in the original version of the paper. The revised manuscript now includes the following added paragraph in the Results Section.

It is significant that the proposed approach is designed specifically to detect (using only standard SCADA data, which are usually 10-minute averaged) the possible heat generated from an initial failure mode, such as the initiation or propagation of the crack, friction, electrical discharge and other failure modes associated with heat release. These types of failure typically result in a gradual and sustained increase in temperature (while they evolve), rather than sudden spikes or drops, which makes them detectable even with low sampling rates, as temperature variables have a low dynamic and

still contain the information of the fault after being 10-minute averaged. With respect to the use of the weekly average, it is intended to reduce false positives by smoothing out transient fluctuations in the data that are not indicative of actual anomalies. Although this averaging may limit the resolution of the approach, as it could smooth out subtle changes in the data that could be indicative of early-stage anomalies, this trade-off is necessary to minimize false alarms and ensure practical utility of the methodology (and avoid alarm fatigue).

The data show a clear seasonal component. The question arises why this was not removed by decomposition, especially since the model is only based on individual data sets. After peaks (Figure 11), the signal drops sharply again for longer periods. What is the reason for this? Since the level seems to be significantly lower in the period from May to November, it is questionable whether incipient damage could be detected here at all.

**Author's reply**: Regarding your question about the seasonal component in the data, we agree that this is an important consideration, and we did take steps to address it in our analysis. Specifically, we subtracted the ambient temperature to all variables related to temperature and used a rolling window approach to train our model on a subset of the data, which helped to capture the seasonality and other temporal patterns in the data. We acknowledge that there are other methods for removing seasonality from time series data, such as seasonal decomposition, and we will consider these approaches in our future work.

Finally, thank you very much for your comment regarding the after peaks signal dropping sharply again for longer periods. The reason for this has been explained in the revised manuscript, in the Results Section, where the following paragraph has been added.

Note that after peaks (Figure 11, WT5), the signal drops sharply again for a long period. This is because the heat created from an initial failure mode (heating from an initial crack, friction, wear,...) is detected by the methodology, but its appearance is not continuous over time until the final breakdown. In contrast, when the failure mode advances, for example, when a crack propagates, the generated heat appears. When the crack remains still, no further heat is generated; thus, the alarm is set off. However, cracks are already present and can advance at any time, leading to the possible failure of the component. Thus, in this methodology, whenever the alarm is on (even when it is set off after a few weeks), it is highly recommended to check the specific WT.

Overall, despite the previously mentioned criticisms, the work is important because it helps to further advance the topic of machine learning and discuss the benefits and methods.

**Author's reply**: We are pleased to hear that you recognize the importance of our work in advancing the topic of machine learning for fault detection in wind turbines.

The anomalies also do not allow any statement on the type of damage present and the severity of the error. At the same time, there is no statement about the historical data and any false alarms. For a scientific consideration, a hit rate AND an error rate must be given.

**Author's reply**: We acknowledge the issue regarding the capability of the model in detecting the type of damage present and the severity of the error. We believe that further developments could be pursued in this direction, for instance, by incorporating high-sampling rate data and/or additional sensors to improve the precision of the fault location. However, we also recognize that this may come at the cost of increased complexity and expense, that we are trying to avoid in our contribution. While our method may not provide detailed information on the type of damage or its severity, it can still provide valuable insights into the system's performance and indicate the need for further investigation or maintenance actions. As not being able to adequately locating the fault is a clear limitation of the proposed methodology, in the revised manuscript we added the following paragraph in the Conclusions Section (highlighted in blue color as Reviewer 1 also commented on this issue).

> While the main bearing temperature was found to be a suitable indicator for detecting faults in wind turbines, as also stated in a recent paper by Murgia et al. (2023), another limitation of the proposed approach is that it cannot precisely locate the fault or its severity. Further developments could be pursued in this direction, for instance, by incorporating high-sampling data and/or additional sensors to improve the precision of the fault location. However, this may come at the cost of increased complexity and expense, which is trying to be avoided in this work where the main objective is to contribute a cost-effective solution where all variables used are readily available in all industrial-size wind farms (both older and newer).

In regard to the proposal to incorporate a hit and error rate, we thank the reviewer for taking this into our attention. The revised manuscript incorporates the following paragraph in the Results Section.

> In summary, 18 wind turbines were examined, of which 16 were considered healthy and correctly classified as such. One turbine

had the fault of interest and was correctly classified as faulty. Another turbine had a fault (that was not the fault of interest) and was classified as faulty, which could be considered a false alarm. However, in practice, the fact that an alarm was raised for a fault in a different component could still be useful, as it indicates the need for maintenance or further inspection. Therefore, in addition to the hit rate and error rate, the practical implications of false alarms should also be taken into account.

The metrics in Table 4 need to be explained.

**Author's reply**: Thanks for bringing this issue to our attention. The metrics in Table 4 are now detailed in the revised manuscript.

Line 133 and Figure 3: It is not described what kind of damage is typically involved here (lack of lubrication, wear, pitting, ...). Each defect should produce certain characteristics in the measured variables.

**Author's reply**: We agree that each type of damage could produce unique characteristics in the measured variables. However, in our case, the only information available regarding the fault is the work order information stating "Replacing Main Bearing." This limited information makes it difficult to determine the exact type of damage involved.

The structural, linguistic and graphic quality of the publication is very good. The work is clearly structured and the tables, graphs and pictures are easily recognizable and informative.

**Author's reply**: We believe that clear presentation and effective communication are essential in scientific publications, and we are delighted that our work meets these standards. We appreciate your review and will continue to strive for high-quality presentation and communication in our future work.

**Technical corrections**
Figure 16 with WT16 should be placed closer to line 326 where it is addressed.

**Author's reply**: We appreciate your attention to detail and your effort to provide constructive feedback that can improve the readability and clarity of our work. We placed the figure closer to line 326 in the revised manuscript.

Finally, we would like to thank the reviewer for the valuable feedback and the time to review the paper.

---

## Author Comment (AC3)

**Anomaly-Based Fault Detection in Wind Turbine Main Bearings**

Lorena Campoverde-Vilela, María del Cisne Feijóo, Yolanda Vidal, José Sampietro, and Christian Tutivén

**Response to reviewers**

**General comments of the authors**

Dear Editor and the Reviewers,

We sincerely thank you for your constructive comments. Under the reviewers' comments and suggestions, the manuscript has been significantly strengthened both in contents and clarity. Below, you can see the changes that we made in response to each reviewer's comment.

The editor and reviewers found the paper of interest, yet they felt that several issues needed to be improved and clarified before the paper could be accepted for publication. In the revised manuscript:

- The changes made in response to Reviewer 1 are marked in blue.

- The changes made in response to Reviewer 2 are marked in red.

- The changes made in response to Reviewer 3 are marked in brown.

**Reviewer 3**

The authors developed a method to detect wind turbine main bearing failures in an early stage, hence the work fits good to the scope of the journal. They used an anomaly detector based on principal component analysis to detect failures of a main bearing with the help of SCADA data. To train the model and to evaluate the results they used the data of 18 turbines.

The structure is clear, and the steps are described in detail. The overall quality is very good, some minor suggestions in the technical comments may help to improve it a little bit.

It is perfectly fine that the focus is on the model, the selection of data and data processing. Nevertheless, in my opinion the technical background could be highlighted more.

**Author's reply**: Thank you for taking the time to review our paper. We are glad to hear that you find our work relevant to the scope of the journal

and that the structure and description of the steps are clear. We also appreciate your positive comments on the overall quality of our work.

Regarding your suggestion to highlight the technical background more, we agree that it is an important aspect of our work. We revised the manuscript to better explain the underlying concepts and methodologies.

We acknowledge your suggestions for discussing certain aspects of the work in greater detail. We will address them in this point-by-point answer to the suggestions given for improvement.

**Special comments**

E.g. the work of Carrol et. al. (DOI: 10.1002/we.1887) could help to underline the importance to prevent failures and downtimes.

**Author's reply**: Thanks for bringing this issue to our attention. In our revised manuscript, we included in the Introduction Section a discussion of the importance of early detection of wind turbine main bearing failures in reducing downtime and maintenance costs. We referenced the work of Carrol et al. as an example of related research in the field. In particular, the following paragraph has been added to the revised manuscript.

> Early detection of main bearing failures of wind turbines is crucial to guarantee the reliability of the element, as well as a safer and more efficient operation in wind farms. The main bearing is one of the most critical components in a WT, and a failure in it can cause significant damage to other components, such as the gearbox, generator, and blades, and result in downtime and expensive repairs, see Carrol et al. (2016). Early detection of main bearing failures enables predictive maintenance, giving maintenance crews time to plan and schedule repairs during low wind periods, minimizing the impact on energy production.

To give technical details of a WT is not necessary. In my opinion the power curve in figure 1 does not give any contribution to this work. The lines from 97 to 102 could be deleted. Here a reference to other publications like Hansen would be possible as well. However, the authors do not give information about main bearings. Possible questions are: Which kind of suspension do the turbines have? Why do I need a bearing and what are possible bearing types? Maybe its not necessary to explain it in detail, but at least a reference would be welcome (Wenske 2022 DOI: 10.1049/PBPO142F or Hau . . . .). A cross reference to figure 2 can be done, too.

**Author's reply**: We appreciate your suggestion that the power curve may not be necessary for this work and agree that a reference to other publications may be more appropriate. Therefore, we removed the lines from 97 to 102 of the original manuscript and referenced Hansen's work in our revised manuscript.

We understand the importance of streamlining the manuscript to focus on the core contributions of our research and appreciate your input in this matter. In particular, the following paragraph was added in Section 2.

> Technical details of the wind turbines under study are out of the scope for the analysis presented in this paper. However, it should be noted that wind turbine design and operation can impact the performance of fault detection methods. The book of Hansen (2015), on the aerodynamics of wind turbines, provides a comprehensive overview of wind turbine design and operation, including factors that can impact the accuracy of fault detection methods. Therefore, we encourage readers who are interested in the technical details of wind turbine design to refer to this resource.

In regard to the main bearing given information, we appreciate your suggestion that additional information or a reference to relevant literature would be beneficial. In response to your feedback, the following paragraph has been added in Section 2.

> Regarding the drivetrain configuration, three-point and four-point suspensions, which refer to one or two main bearings, respectively, are the most common wind turbine drivetrain architectures. In the three-point suspension configuration, which is the one used in the wind farm under study, the rotor is rigidly connected to the main shaft, which is supported by a single main bearing near the rotor. A shrink disk typically connects the downwind side of the shaft to the low-speed stage of the gearbox. The gearbox is supported by two torque arms that are connected to the bedplate elastically. These two torque arms, along with the single main bearing, provide a total of three points of support. Furthermore, there are different types of state-of-the-art main bearings, as fully explained in Wenske (2022). In particular, the turbines of this park are equipped with the so-called spherical roller bearing (SRB) type. SRBs are characterized by their outer raceways being a portion of a sphere. The rollers, in turn, are shaped so that they closely conform to the inner and outer raceways. This results in a bearing that is internally self-aligning and has a high radial load carrying capacity, please see Hart et al. (2019) for a more detailed explanation.

There are plenty of possible bearing damages (fatigue, wear cracks... they can occur at the rings, raceways, rollers or at the cage) which can have an effect on the bearing lifetime. This is not considered. Here I can recommend e.g. the work of Hard like DOI: 10.1002/we.2386. As a reference about bearing damages e.g. the work from Harris and Kotzalas could be used. The fact that just one main bearing failure occurs in the data, may raise the question if

other main bearing failures can be detected. At least in the discussion or in the outlook I would expect a discussion on that.

**Author's reply**: Thank you for your comments on the potential different types of bearing damage and locations. In response to your feedback, we included a brief discussion in the manuscript on the possible bearing damage modes, such as fatigue, wear, and cracks, and their impact on the bearing lifetime. We referenced the works of Hard (DOI: 10.1002/we.2386) and Harris and Kotzalas to provide additional information on these topics. In particular, the following paragraph has been added to the Introduction Section.

> Bearing damage in wind turbines can occur in different locations, including the rings, raceways, rollers, and cage. The most common types of bearing damage are related to heat release, which can result from friction, wear, and cracks, see Harries et al. (2006). All of these damage modes can significantly impact the lifetime of the bearing, which in turn can cause significant downtime and maintenance costs. Early detection of bearing damage through monitoring and detection of heat release can allow for timely repairs and maintenance, minimizing the impact on the bearing and other components, and reducing downtime and maintenance costs. The methodology proposed in this work aims to detect heat release in the bearings, allowing for early detection and diagnosis of potential bearing damage.

Furthermore, in the Results Section the following paragraph has also been added (highlighted in red color as Reviewer 2 also commented on this issue).

> It is significant that the proposed approach is designed specifically to detect (using only standard SCADA data, which are usually 10-minute averaged) the possible heat generated from an initial failure mode, such as the initiation or propagation of the crack, friction, electrical discharge and other failure modes associated with heat release. These types of failure typically result in a gradual and sustained increase in temperature (while they evolve), rather than sudden spikes or drops, which makes them detectable even with low sampling rates, as temperature variables have a low dynamic and still contain the information of the fault after being 10-minute averaged.

Regarding the occurrence of only one main bearing failure in our dataset, we acknowledge that this may raise questions about the generalizability of our approach in detecting other main bearing failures. We addressed this concern in the revised manuscript by including the following paragraph in the Conclusions Section.

> The results demonstrate that the stated approach is effective in detecting a main bearing fault that resulted in a significant increase

in temperature. Although only one failure was available in the investigated wind park data, which is insufficient for statistical analysis, any bearing fault leading to heat release might be detectable by the proposed strategy. However, to more extensively investigate the performance of the model, it is necessary to apply the model to other wind parks with main bearing failure issues. Therefore, future work will test the model on a larger dataset to assess its performance in different scenarios and draw more generalizable conclusions.

The mentioned counteractions to prevent a bearing failure after detection stay very vague.

**Author's reply**: Thank you for your comment about the counteractions to prevent a bearing failure after detection. In response to your comment, the following paragraph was added (in red color as this issue was also commented by Reviewer 2) to the Results Section of the revised manuscript.

Note that after peaks (Figure 11, WT5), the signal drops sharply again for a long period. This is because the heat created from an initial failure mode (heating from an initial crack, friction, wear,...) is detected by the methodology, but its appearance is not continuous over time until the final breakdown. In contrast, when the failure mode advances, for example, when a crack propagates, the generated heat appears. When the crack remains still, no further heat is generated; thus, the alarm is set off. However, cracks are already present and can advance at any time, leading to the possible failure of the component. Thus, in this methodology, whenever the alarm is on (even when it is set off after a few weeks), it is highly recommended to check the specific WT.

**Technical comments**
In Table5 and figure 5 units are missing.

**Author's reply**: Thank you for pointing out that units are missing in Table 5 and Figure 5. We apologize for this oversight. We ensured that the missing units are included in the revised manuscript.

In figures 11, 12, and 13 a same y-axis scale would make it easier to compare the individual turbines.

**Author's reply**: We appreciate your feedback and understand your suggestion of using a consistent $y$-axis scale for comparison. However, we decided to

use different $y$-axis scales for each figure as the models were trained on data with different characteristics and had different ranges of normal and faulty data. Using a consistent $y$-axis scale could potentially cause misleading visual comparisons of the data.

We have, however, kept the threshold value in the same position in each figure to provide a clear comparison between the actual and predicted values of each turbine, making it easier to see whether a turbine is operating normally or has a fault. The position of the threshold value is independent of the $y$-axis scale and is used to classify the data points, making it a crucial reference for the reader.

We hope that this explanation clarifies our reasoning for using different y-axis scales but maintaining the same position for the threshold value in each figure. If you have any further suggestions or comments, please let us know.

To reduce the number of plots it could be a good idea to summarize a few turbines in one plot. Different colors could be used.

**Author's reply**: Thank you for your suggestion to summarize multiple turbines in one plot and use different colors. While we appreciate your idea, we would like to keep one plot per wind turbine for the sake of clarity and ease of interpretation. Our aim is to provide a clear and detailed presentation of each turbine's performance, and we believe that this would be better achieved through individual plots. By doing so, readers can easily compare the performance of each turbine and identify any differences or patterns that may arise.

Sometimes shorter sentences would increase the legibility. As one example would separate the sentence (in line 326) after the first date.

**Author's reply**: Thank you for your suggestion to use shorter sentences for improved legibility, and for highlighting the sentence in line 326 as an example. We will separate the proposed sentence and also thoroughly review the entire manuscript to identify and address similar issues to ensure that the text is presented in a clear and concise manner.

Finally, we would like to thank the reviewer for the valuable feedback and the time to review the paper.